# Deciphering “Immaturity-Stemness” in Human Epidermal Stem Cells at the Levels of Protein-Coding and Non-Coding Genomes: A Prospective Computational Approach

**DOI:** 10.3390/ijms25063353

**Published:** 2024-03-15

**Authors:** Tatiana Vinasco-Sandoval, Gilles Lemaître, Pascal Soularue, Michèle T. Martin, Nicolas O. Fortunel

**Affiliations:** 1Université Paris-Saclay, 91190 Gif-sur-Yvette, France; gloria.vinasco-sandoval@cea.fr (T.V.-S.); pascal.soularue@cea.fr (P.S.); michele.martin@cea.fr (M.T.M.); 2Commissariat à l’Energie Atomique et aux Energies Alternatives (CEA), Institut de Radiobiologie Cellulaire et Moléculaire (iRCM), Laboratoire de Génomique et Radiobiologie de la Kératinopoïèse (LGRK), 2 Rue Gaston Crémieux, 91000 Evry, France; gilles.lemaitre@cea.fr; 3CEA, Institut de Biologie François Jacob (IBFJ), 92265 Fontenay-aux-Roses, France; 4Université Paris-Saclay, Univ Evry, 91000 Evry, France

**Keywords:** skin, stem cells, stemness, bioinformatics, transcriptome, coding genes, non-coding genes

## Abstract

The epidermis hosts populations of epithelial stem cells endowed with well-documented renewal and regenerative functions. This tissue thus constitutes a model for exploring the molecular characteristics of stem cells, which remain to date partially characterized at the molecular level in human skin. Our group has investigated the regulatory functions of the KLF4/TGFB1 and the MAD4/MAX/MYC signaling pathways in the control of the immaturity-stemness versus differentiation fate of keratinocyte stem and precursor cells from human interfollicular epidermis. We described that down-modulation of either *KLF4* or *MXD4*/MAD4 using RNA interference tools promoted an augmented stemness cellular status; an effect which was associated with significant transcriptional changes, as assessed by RNA-sequencing. Here, we have implemented a computational approach aimed at integrating the level of the coding genome, comprising the transcripts encoding conventional proteins, and the non-coding genome, with a focus on long non-coding RNAs (lncRNAs). In addition, datasets of micro-RNAs (miRNAs) with validated functions were interrogated in view of identifying miRNAs that could make the link between protein-coding and non-coding transcripts. Putative regulons comprising both coding and long non-coding transcripts were built, which are expected to contain original pro-stemness candidate effectors available for functional validation approaches. In summary, interpretation of our basic functional data together with in silico biomodeling gave rise to a prospective picture of the complex constellation of transcripts regulating the keratinocyte stemness status.

## 1. Introduction

The scientific challenge of deciphering the molecular components that contribute to the regulation of the stemness status was particularly highlighted in the 2000s [1,2,3,4,5]. However, this concept remains to this day associated with unresolved fundamental questions. In particular, the identification of molecular species from the non-coding genome, including the class of long non-coding RNAs (lncRNAs), brings a level of complexity to the achievement of this objective [6]. Indeed, a flow of data leads to the observation that integrating the contributions of the non-coding genome into the regulatory networks governed by the protein-coding genome—i.e., encoding the proteome—is necessary to address in a relevant way the processes of physiological regulation and pathophysiological disturbances [7]. The choice of addressing these questions in humans brings a parameter of difficulty, compared to rodents, where transgenic models can be developed for functional and mechanistic studies. 

Epidermal stem cells, which are also named keratinocyte stem cells from human interfollicular epidermis, constitute a relevant cellular model to decipher the stemness in a human tissue. Indeed, their functional characteristics are well established, according to two major clinical advances obtained in cutaneous biotherapies. The first one is the demonstration that the keratinocyte stem cells hosted within the human interfollicular epidermis are endowed with a regenerative potential that enables skin substitute bioengineering and patient grafting on large body surfaces affected by irreversible third-degree burns [8,9]. The second one is the long-term correction of a genodermatosis (i.e., junctional epidermolysis bullosa) via stable viral vector-driven insertion of a therapeutic transgene in the genome of keratinocyte stem cells, in the context of a gene therapy clinical trial [10,11,12]. The cellular material corresponding to the clonal progeny of keratinocyte stem cells can be cultured and maintained ex vivo, and is functionally defined as “holoclones”. Keratinocyte holoclones are endowed with the capacity for self-renewal, resulting in a growth potential exceeding 100 population doublings [13,14]. Experimentally, their regenerative potential can be assessed in vitro by their efficient capacity for three-dimensional epidermis reconstruction [14,15,16], and in vivo by the xenografting of skin reconstructs in immunocompromised recipient animals [17,18,19]. In summary, the cellular model of holoclone keratinocytes has the advantage of connecting basic researches aimed at acquiring knowledge on epidermal stem cells and the clinical side of their uses.

Our group has used holoclone keratinocytes as a cellular model to investigate the function of two candidate transcription factors in the regulation of the immaturity/stemness versus differentiation fate decision in epidermal stem and precursor cells. A functional genomics approach based on stable lentiviral vector-mediated transduction of short-hairpin interfering RNAs (shRNAs) was set up to compare the molecular characteristics and functionalities of transcription factor “wild-type” and “knockdown” holoclone keratinocytes. Using this principle, we first documented that the expression level of “Kruppel-like factor 4” (*KLF4*) constitutes a control mechanism of this balance. A low *KLF4* level drives cells towards self-renewal and expansion, whereas a higher *KLF4* expression orientates cells towards reduction of stemness-immaturity and entry into differentiation [19]. We next documented an equivalent role for our second candidate “MAX Dimerization Protein 4” (*MXD4*/MAD4), as its low expression level also promoted immaturity-stemness and self-renewal [16]. For both transcription factors, we characterized the transcriptome profiles of “wild-type” and “knock-down” holoclone keratinocytes by RNA-sequencing (RNA-seq). In first intention, transcriptome modulation was investigated at the level of the proteome-coding genome, but the results of the two studies were not cross-analyzed.

Here, we implemented a computational approach aimed at increasing the scope of the bioanalysis by integrating the data corresponding to the proteome-coding transcriptome and those corresponding to the non-coding transcriptome with a focus on lncRNAs; in view of identifying crosstalk between the signaling modulated by *KLF4* and *MXD4*/MAD4.

## 2. Results

### 2.1. Cellular Model

As previously described in our original articles [16,19], the cellular model of human holoclone keratinocytes was used to decipher the molecular network responsible for the regulation of human epidermal stem and precursor cell fate. These cells correspond to the clonal progeny of single keratinocyte stem cells. They have been functionally characterized ex vivo by their growth potential, exceeding 100 population doublings in long-term culture, and their capacity for three-dimensional (3D) epidermis reconstruction. Additionally, holoclone keratinocytes have the potential for long-term in vivo grafting. The functional genomics approach designed to explore the regulatory roles of KLF4 and *MXD4*/MAD4 was based on the generation of stable targeted knock-down (KD) cellular contexts. These contexts were then used for comparative studies of KD cells versus their wild-type (WT) equivalent. Lentiviral vectors driving expression of shRNAs directed against the *KLF4* or *MXD4* transcripts were used to transduce holoclone keratinocytes and obtain stable KD cells. Comparisons were performed versus cells transduced with a control vector. The *KLF4^KD^* and the *MXD4^KD^* strategies converged to promote an augmented ex vivo cellular expansion associated with improved maintenance of stem and precursor cell clone-forming efficiency, together with preservation of potential for epidermis generation [16,19] (Figure 1). Accordingly, the transcriptome datasets modulated in response to *KLF4* and/or *MXD4* KD constituted a relevant material for analyzing the transcriptional networks controlling immaturity-stemness and self-renewal in human keratinocyte stem and precursor cells. 

### 2.2. RNA-Seq Dataset Processing

The raw RNA-seq datasets corresponding to *KLF4^WT^* and *KLF4^KD^* keratinocytes are available in the Gene Expression Omnibus (GEO) database under accession number GSE111786. Those corresponding to *MXD4^WT^* and *MXD4^KD^* cells are available via the accession code GSE202700. Each dataset was based on three biological replicates for both wild-type (WT) and knock-down (KD) cellular contexts. A total of 17 RNA-seq libraries including biological and technical replicates were sequenced for the *KLF4* study, and six libraries were sequenced for the *MXD4* study (Appendix A). Batch effects were evaluated and corrected in the *KLF4* dataset to group the technical replicates corresponding to each biological replicate. This resulted in a count matrix with unique values for each biological replicate. *KLF4* library sizes varied between 30 and 100 million reads per sample (Appendix A), whereas *MXD4* library sizes were more homogeneous, with an average of 62.7 million reads per sample (Appendix A). All samples from both *KLF4* and *MXD4* datasets underwent extensive quality controls, including filtering lowly expressed reads, multidimensional reduction methods (MDS), and TMM (trimmed mean of M-values) normalization to adjust reads to library size. After filtering reads detected at too low a level (under 10 reads), we obtained 18,003 and 21,315 expressed transcripts for the *KLF4* and the *MXD4* datasets, respectively. Comparable percentages of protein-coding transcripts (pcRNAs) and lncRNAs were found for both datasets; these were around 70–78% and 15–20%, respectively (Appendix A). 

### 2.3. Protein-Coding Transcripts (pcRNAs) Responsive to KLF4 Knock-Down

In the *KLF4* study, RNA-seq detected a total of 40,077 transcript sequences with at least one read. From these sequences, transcripts detected with at least 10 reads in three samples were kept for the next steps of the analysis. This selection led to the identification of 18,003 expressed transcripts, comprising 94.9% of pcRNAs. Then, transcripts differentially expressed (DE) in *KLF4^WT^* and *KLF4^KD^* keratinocytes were extracted according to false discovery rate (FDR) < 0.05 and absolute fold-change (FC) > 1.5. A total of 2712 DE transcripts were identified, which accounted for 15% of the total expressed transcripts. Among these, 2311 were pcRNAs (85.3% of DE transcripts). As expected, the *KLF4* transcript was found to be down-modulated in all three samples of anti-*KLF4* shRNA-transduced cells. The DE transcriptional signature consisted of 995 up-modulated pcRNAs and 1316 down-modulated pcRNAs when comparing *KLF4^KD^* versus *KLF4^WT^* keratinocytes. (Figure 2A and Table 1). The volcano plot representation shown in Figure 2B highlighted the robustness of the KD principle, since −log_10_P rose up to 6 for several DE pcRNAs. Increasing the selection threshold to FC > 2 (with same FDR < 0.05) still identified a large set of DE transcripts, which comprised 490 up-modulated pcRNAs and 377 down-modulated pcRNAs. The top most significant DE pcRNAs are listed in Table 2. 

### 2.4. Long Non-Coding Transcripts (lncRNAs) Responsive to KLF4 Knock-Down

In the RNA-seq dataset of the *KLF4* study, sequences assigned to lncRNAs detected by at least one read represented 23.1% of total transcripts. This percentage dropped to 5.1% when considering transcript detection corresponding to at least 10 reads in three samples. Among the 18,003 total expressed transcripts detected using this threshold, 918 lncRNAs were identified. We then extracted DE lncRNAs in *KLF4^WT^* and *KLF4^KD^* keratinocytes, according to FDR < 0.05 and FC > 1.5. From the total of 2712 DE identified transcripts, 401 sequences corresponded to lncRNAs. This DE transcriptional signature comprised 175 up-modulated lncRNAs and 226 down-modulated lncRNAs [*KLF4^KD^* versus *KLF4^WT^* keratinocytes] (Figure 2C and Table 1). From these, a total of 329 DE lncRNAs (136 up-modulated and 193 down-modulated) were still identified when the absolute FC threshold was increased to 2 (with same FDR < 0.05). The most significant DE lncRNAs identified in the *KLF4* study are listed in Table 3. 

### 2.5. Protein-Coding Transcripts (pcRNAs) Responsive to MXD4 Knock-Down

A similar analysis to that described for the *KLF4* transcriptome dataset was conducted on the RNA-seq datasets from the *MXD4* study. A total of 45,766 transcripts were detected with at least one read. However, only transcripts with detection levels of 10 reads or more, corresponding to 21,315 transcripts, were considered for the subsequent analysis. From these, 90.4% were pcRNAs. Extraction of DE transcripts in *MXD4^WT^* and *MXD4^KD^* keratinocytes according to FDR < 0.05 and FC > 1.5 identified a set of 6664 transcripts (31.2% of total expressed transcripts) (Figure 3A), among which 5200 were pcRNAs (78.0% of DE transcripts). As expected, the *MXD4* transcript was down-modulated in anti-*MXD4* shRNA-transduced cells (Figure 3A). The DE transcriptional signature comprised 3151 up-modulated pcRNAs and 2049 down-modulated pcRNAs [*MXD4^KD^* versus *MXD4^WT^* keratinocytes] (Table 4). As observed for the *KLF4* datasets, the volcano plot representation of *MXD4* data (Figure 3B) highlighted the confidence of the KD principle, since −log_10_P rose up to 6 for several modulated RNAs. Using FC > 2 and FDR < 0.05, we still found a large number of DE transcripts, with 1218 pcRNAs up-modulated and 1554 pcRNAs down-modulated in *MXD4^KD^* cells (Figure 3B). The top most significant DE pcRNAs are listed in Table 5. 

### 2.6. Long Non-Coding Transcripts (lncRNAs) Responsive to MXD4 Knock-Down 

Analysis of the non-coding *MXD4* RNA-seq dataset assigned 28% of the sequences detected by at least one read to lncRNAs. This percentage dropped to 9.6% when the threshold of 10 reads was applied to consider only significant transcript detections, which identified 2046 lncRNAs among the 21,315 total expressed transcripts. Then, extraction of modulated transcripts according to FDR < 0.05 and FC > 1.5 identified 1464 DE lncRNAs, which corresponded to 21.9% of the 6664 total DE transcripts. This DE transcriptional signature comprised 220 up-modulated lncRNAs and 1244 down-modulated lncRNAs [*MXD4^KD^* versus *MXD4^WT^* keratinocytes] (Figure 3C and Table 4). From these, a total of 1273 DE lncRNAs (151 up-modulated and 1122 down-modulated) were still identified when the absolute FC threshold was increased to 2 (with same FDR < 0.05). The top most significant DE lncRNAs identified in the *MXD4* study are listed in Table 6.

### 2.7. Functional Enrichment Analysis of DE pcRNAs Identified in the KLF4 and MXD4 Datasets by Over-Representation Analysis (ORA)

In order to identify transcriptional pathways modulated in association with the cellular character of augmented immaturity-stemness status, a functional enrichment analysis of DE pcRNAs was performed. The DE transcript sets for the *KLF4* and *MXD4* studies, consisting of 2311 and 5200 pcRNAs, respectively, underwent Over-Representation Analysis (ORA) using seven gene set databases: KEGG, GO Biological processes, MSigDB Hall-marks, Reactome, ChEA ChipX Experiment Analysis, Bioplanet, and Tabula Muris. ORA is a statistical method used to determine whether transcripts from a priori defined set are over-represented in a particular transcript selection or signature. Numerous pathways were highlighted for each dataset (Appendix A). Interestingly, convergences were found between the transcriptional pathways modulated in the *KLF4* and the *MXD4* studies (Figure 4). In particular, the Bioplanet and GO Biological processes databases revealed modulation of transcripts involved in growth factor signaling. These included the epidermal growth factor (EGF), the brain-derived neurotrophic factor (BDNF), and the wingless-int1 (WNT) pathways. Of note, functional enrichment analysis using the ChipX Experiment Analysis database pointed on modulated transcripts that are under the control of transcription factors with documented regulatory functions in epidermal keratinocytes, such as the tumor protein p63 (TP63), the forkhead box protein M1 (FOXM1), and the estrogen receptors 1 and 2 (ESR1 and ESR2). 

### 2.8. Functional Enrichment of pcRNAs Identified in KLF4 and MXD4 Datasets by Unsupervised Gene Set Enrichment Analysis (GSEA)

Gene set enrichment analysis (GSEA) is a rank-based computational approach that determines whether predefined groups of transcripts show statistically significant, concordant differences between two biological conditions. We used GSEA as an alternative method to ORA, in order to strengthen our analysis. GSEA was performed using MSigDB Hallmarks, KEGG, Reactome, Immunologic Signature, and GO-based gene sets and the entire normalized expression signals of pcRNAs identified in the *KLF4* and in the *MXD4* studies, which, respectively, comprised 15,228 and 18,189 transcripts. Focusing on KEGG-based functional exploration, the lists of transcripts from the two studies were compared to 160 KEGG pathway-associated gene sets, in order to view the top 50 most significant pathways modulated in response to *KLF4* or *MXD4* KD. Notably, 50% of them were common to the two datasets, thus documenting convergence in the transcriptional networks modulated in the two KD contexts. Shared pathways represented with positively co-modulated transcripts (up-modulated in *KLF4^KD^* and *MXD4^KD^*) concerned cell-cycle (8 of the 50 most significant KEGG gene sets), amino-acid metabolism (8 KEGG gene sets), fatty-acid metabolism (4 KEGG gene sets), and sugar biosynthesis (3 KEGG gene sets). In contrast, shared pathways represented with negatively co-modulated transcripts (down-modulated in *KLF4^KD^* and *MXD4^KD^*) concerned cell adhesion and cell–cell junctions (12 KEGG gene sets), and signaling by cytokines and growth factors (12 KEGG gene sets). As examples, most of the transcripts assigned to the KEGG pathways related to cell cycle (Figure 5A) and to glycolysis–gluconeogenesis (Figure 5B), were detected as up-modulated both in the *KLF4^KD^* and in the *MXD4^KD^* contexts. In contrast, most of the transcripts assigned to the KEGG pathways related to extracellular matrix and receptor interactions (Figure 5C), and to transforming growth factor-beta (TGFB) signaling (Figure 5D), were found down-modulated in the two cellular contexts. 

### 2.9. Shared Differentially Expressed Transcripts Identified in Both Analyses

To identify all transcripts modulated in both cellular models, DE transcripts identified independently in each analysis were compared (2712 sequences for the *KLF4* and 6664 for the *MXD4* study). A set of 1115 DE shared transcripts (FDR < 0.05 and FC > 2) was identified (Figure 6A,C). Repartition of pcRNA sequences indicates that over the 6555 total DE sequences, 4244 were specific to the *MXD4* model, 1355 to the *KLF4* model, and 956 were modulated in both models (Figure 6A). Regarding lncRNA sequences, from a total of 1706 DE sequences, 1305 were specific to the *MXD4* model, 242 to the *KLF4* model, and 159 were modulated in both models (Figure 6C). 

Concerning shared DE pcRNAs (956 transcripts) (Figure 6B), 175 were found down-modulated in the *KLF4^KD^* and up-modulated in the *MXD4^KD^* cellular models, 268 were found up-modulated in the *KLF4^KD^* and down-modulated in the *MXD4^KD^* models; whereas 322 were identified in both cellular models as down-modulated (Figure 6B, dark blue dots) and 191 as up-modulated (Figure 6B, red dots). The same analysis was performed for shared DE lncRNAs (Figure 6D). Over the 159 common DE lncRNAs (Appendix A), 21 were found down-modulated in the *KLF4^KD^* and up-modulated in the *MXD4^KD^* cellular models, 43 were found up-modulated in the *KLF4^KD^* and down-modulated in the *MXD4^KD^* models; whereas 75 were identified as down-modulated (Figure 6D, dark blue dots) and 20 were identified as up-modulated (Figure 6D, red dots) in both cellular models. 

### 2.10. Functional Enrichment Analysis of DE pcRNAs Common to the KLF4^KD^ and MXD4^KD^ Models by Over-Representation Analysis (ORA)

For this analysis, lncRNAs were not included as there is currently no database available for functional annotation on this category of transcripts. ORA was performed on 956 DE pcRNAs common to *KLF4* and *MXD4* studies using the seven databases: KEGG, GO Biological processes, MSigDB Hallmarks, Reactome, ChEA ChipX experiment analysis, Bioplanet, and Tabula Muris. This analysis was highly convergent with that performed by GSEA as several pathways were identified independently using the two methods (Figure 7). Pathways down-modulated in both *KLF4^KD^* and *MXD4^KD^* keratinocytes comprised transcript groups related to extracellular matrix organization (Figure 7A,D), cell adhesion, growth factor signaling and notably transforming growth factor-beta/bone morphogenic protein (TGFB/BMP) signaling network (Figure 7B). On the contrary, cell cycle-related transcripts were pointed out as up-modulated in both *KLF4^KD^* and *MXD4^KD^* cells. Specificities in the regulatory networks affected by *KLF4* or by *MXD4* expression levels are highlighted by the presence of pathways significantly modulated in one KD context, and not in the second; for example, the interferon signaling pathway that was only found up-modulated in *KLF4^KD^* cells (Figure 7C). Notably, regulators with a documented pro-stemness function in the keratinocyte lineage such as *FOXM1* were found within the transcripts up-modulated in both *KLF4^KD^* and *MXD4^KD^* contexts (Appendix A), which is in accordance with our working model proposing that promotion of immaturity-stemness is obtained by down-modulating *KLF4* and/or *MXD4* expression. Finally, our transcriptomic signature of up-modulated transcripts, which was defined by studying human cells, exhibited a strong convergence with the transcript set designed as “stem cell of epidermis” from the murine single-cell transcriptome database Tabula Muris (Appendix A). These analysis result in a selected signature of 218 shared DE pcRNAs (Appendix A).

### 2.11. Prospective Definition of Regulons Comprising lncRNAs and DE pcRNAs Common to KLF4^KD^ and MXD4^KD^ Cells Using Transcriptional Regulatory Networks (TRNs)

Since the pcRNA–lncRNA interaction databases are currently limited, co-expression networks were constructed to describe pairwise relationships between transcripts. Therefore, we explored the meaning of the lncRNA modulations observed in our KD models building lncRNA-centered regulatory networks independently for each model (*KLF4* and *MXD4*). We applied the transcriptional regulatory network (TRN) analysis that consists of two steps: first, a transcriptional network inference (TNI); followed by the transcriptional networks analysis (TNA). TNI was used to establish the association between the expression of each of the 159 lncRNAs commonly identified in the *KLF4* and the *MXD4* datasets, and 1115 potential RNA targets (956 shared DE pcRNAs and 159 shared DE lncRNAs). Network inference analysis resulted in the construction of 159 regulatory units called regulons, a regulatory unit formed by a regulatory element (lncRNA) and its potential regulatory partners (pcRNAs or lncRNAs). These regulons comprised 159 lncRNA and 1100 RNAs for *KLF4^KD^* model, and 159 lncRNAs and 1007 RNAs for *MXD4^KD^* model. 

Then, TNA was applied to test the expression association between the regulons with the ORA selected signature (218 pcRNAs). TNA results in the identification of 132 regulons for *KLF4* model and 107 for *MXD4* model. The intersection of regulons between the two models leads to the identification of 57 regulatory units formed by 57 lncRNAs and 104 pcRNAs. The complete TNR analysis produced interaction networks involving pcRNAs and lncRNAs, their known targets, and the pathways in which the targets are involved. Two main networks were constructed. A first one, based on 47 regulons containing 47 lncRNAs and 82 pcRNAs, was related to the control of the balance between immaturity-stemness and differentiation in the keratinocyte lineage (Figure 8A). A second one, based on 10 regulons containing 10 lncRNAs and 22 pcRNAs, was related to the cell cycle (Figure 8B). Notably, as expected, a regulon related to the TGFB pathway emerged as a component of these networks (Figure 8A). Co-regulation analysis pointed to seven lncRNAs potentially interacting with four pcRNAs (*BAMBI*, *ID4*, *PITX2,* and *TGFB2*) in this pathway.

### 2.12. Weighted Gene Co-Expression Network Analysis (WGCNA) of DE pcRNAs Common to KLF4^KD^ and MXD4^KD^ Cells with lncRNAs

In parallel with the TRN analysis, we applied the method of gene co-expression network analysis (WGCNA) to identify consensus clusters (modules) of highly correlated transcripts common to the *KLF4* and *MXD4* studies. After normalization and checking for outliers in both datasets, transcripts with at least 10 reads in three samples were considered for WGCNA analysis. We thus obtained 49 consensus co-expression modules with transcript numbers ranging from 70 to 4359 (Figure 9). We then considered the cellular contexts (WT or KD) to establish the module–trait relationships within the network. Some modules exhibited significant relationships with the WT or KD (Figure 9). We then filtered each module using our signature of 159 DE lncRNAs and the ORA selected signature of 218 DE pcRNAs identified in common in the *KLF4* and *MXD4* studies. We selected three distinct consensus networks (correlation *p*-value < 0.05) comprising 36 transcripts (18 lncRNAs and 18 pcRNAs), 69 transcripts (16 lncRNAs and 53 pcRNAs), and 20 transcripts (11 lncRNAs and 9 pcRNAs); these are visualized in dark orange, maroon, and yellow modules, respectively (Figure 10A–C).

### 2.13. Prospective Search of Competitive Endogenous pcRNA-miRNA–lncRNA Interaction Networks

Since databases concerning lncRNA biological activities and interactions with pcRNAs are not yet available, the drawing of interaction networks based on functional annotations concerning the two transcript classes was not possible. In order to initiate an interaction map, we took advantage of known interactions of lncRNAs with miRNAs due to documented biological functions of lncRNAs, such as miRNA sponging. Cross analysis of the signatures identified by TRN and WGCNA led to a selection of 14 shared lncRNAs (Table 7). These lncRNAs were submitted to the StarBase ENCORI tool and only five of them (*CD27-AS1*, *DNM3OS*, *MIR503HG*, *PCBP1-AS1*, and *TBX5-AS1*) were listed for having documented interactions with miRNAs. Selection of miRNAs with a target-directed microRNA degradation (TDMD) score > 0.8 resulted in a list of 18 miRNAs. Each lncRNA-associated miRNA was submitted to miRTargetLink, resulting in different independent transcript networks documented by experimental validations. The sum of specific networks was merged to draw a global competitive endogenous pcRNA–miRNA–lncRNA interaction network (Figure 11). Notably, several pcRNAs involved in TGFB signaling, together with other growth factor pathways, were highlighted. Interestingly, *miR-92a-3p*, *miR-25-3p*, which is linked to the lncRNA *TBX5-AS1*; and *miR-15a-5p*, which is linked to the lncRNA *MIR503HG*, are described to interact with *KLF4* transcript. 

## 3. Discussion

Deciphering molecular networks that ensure the regulation of the immaturity-stemness versus differentiation fate is a common challenge in the domain of stem cell biology. In this study, we investigated this issue taking advantage of the model of human stem and precursor cells from human interfollicular epidermis. A sum of transcriptional studies were already conducted using bulk or single cell transcriptome analysis [20] on cellular models in culture [16,19,21,22] or on fresh tissue [23,24]. Of note, the majority of these studies were carried out with a focus on the gene fraction that ensures the proteome coding (i.e., protein-coding RNAs/messenger RNAs). The development of knowledge on epigenetic regulation mechanisms (i.e., chromatin compaction and structure, DNA methylation, acetylation, etc.); and on epigenomic effector classes (i.e., micro-RNAs, circular RNAs, long non-coding RNAs, etc.) is changing our perception of cell fate regulatory mechanisms. In particular, lncRNAs are involved in a sum of regulatory processes, including modulation of protein-coding gene expression [6], and much remains to be discovered regarding their roles and regulatory functions in physiological processes and pathophysiological contexts. Elucidation of complex constellations of regulatory components and effectors will certainly result from the combination of studies addressing complementary levels of information: (1) researches developed for the functional characterization of specific candidates or mechanisms; (2) molecular profiling dedicated to large-scale data acquisition, such as global transcriptomic, proteomic, and/or metabolomics studies; (3) prospective computational approaches aiming at opening avenues for experimental researches, as performed here. 

The possibility of addressing mechanistic questions inherent to the understanding of cellular and tissue functions is conditioned by the availability of relevant experimental models. A difficulty in studies involving adult stem cells is that their immature phenotype can be altered when they are removed from their in vivo niche and placed in an ex vivo culture environment. As illustrated here, human keratinocyte precursor cells driven or not to an augmented immaturity-stemness state by controlling the levels of the transcription factors *KLF4* [19] or *MXD4*/MAD4 [16] was a suitable model for prospective investigation of the molecular networks involved in the regulation of this stem cell-associated character. The biological significance of the *KLF4*- and *MXD4*-based cellular models was supported by data published by different groups, including ours. 

Firstly, a regulon centered on TGFB signaling was identified within the common transcriptional signatures detected as modulated in both models (Figure 12). This can be linked to functional data showing that a fine tuning of TGFB1 signal and response modulates the ex vivo expansion of human keratinocyte precursor cells and their immaturity degree [19,25]. Moreover, TGFB signaling inhibition associated with ROCK signaling inhibition synergistically supports long-term expansion of epithelial precursor cells from prostate, bronchial tissue, and skin [26]. In addition, Activin/TGFB signaling is also involved in the quiescence of hair follicle keratinocytes in mice [27]. Interestingly, the key role of TGFB signaling in the regulation of progenitor cell fate has been also documented in the human hematopoietic system [28]; notably, in the control of cell-cycling and self-renewal [29], and in the equilibrium between immaturity and erythroid differentiation [30]. Secondly, regarding the possible cross-talk between the EGF, BDNF, and WNT pathways (see Figure 12), as suggested by our in silico analysis, no published data are available to directly address this point in interfollicular keratinocytes. It is worth noting that EGFR has been shown to play a critical role in attenuating Wnt/β-catenin signaling during postnatal mouse hair follicle development by directly acting on stem cell-specific markers such as SOX9 and NFATc1 [31]. A comparable mechanism cannot therefore be excluded in interfollicular epidermis keratinocytes. However, there are no available data to confirm the contribution of BDNF in this potential cross-talk among keratinocytes. Finally, the signature defined through the computational approach described in this study exhibited significant convergence with the molecular profile of human keratinocyte holoclones and paraclones established independently (i.e., progeny of stem cells and progeny of less immature progenitors, respectively) [21]. This transcriptional signature identifying holoclones from paraclones (1044 protein-coding (pc) RNAs), and the signatures that we established by comparative profiling of the *KLF4* and the *MXD4* knock-down contexts versus a control keratinocyte precursor state (respectively, 2311 and 5200 pcRNAs), pointed to a list of 118 common pcRNAs. This common transcript set thus represents a relevant basis for understanding the molecular features associated with the keratinocyte immaturity-stemness character. 

This list includes the stem cell-related Forkhead box transcription factor M1 (*FOXM1*) transcript, and various transcripts of the mitogen-activated protein kinases (MAPK), phosphoinositide 3-kinase (PI3K), and ras-related protein 1 (RAP1) signaling pathways. Cell cycle-related transcripts were also present in the three datasets, which is consistent with described signatures of epidermis basal keratinocyte subpopulations obtained by single-cell RNA-seq from human skin [23]. Interestingly, transcripts of the TGFB pathway were found in common in the three datasets. Concerning the soundness of the principle of immaturity-stemness promotion by lowering *KLF4* or *MXD4*, a clear link is observed between transcripts associated with this effect and those found in the transcriptome profile characteristic of holoclones, which correspond to keratinocyte stem cells endowed with the regenerative potential required for long-term graft perennation in clinics [11,12,21]. Of note, some transcript groups and pathways appeared modulated in an opposite manner in our two cellular models (for example, transcripts assigned to “interferon signaling pathways”), which can be interpreted as the result of specificities distinguishing the regulatory networks of KLF4 and *MXD4*/MAD4.

Beside protein-coding transcripts, the revisiting of our datasets detailed in this study, integrating a recent state of knowledge on the different classes of transcripts, opens original perspectives to approach the understanding of the positioning of lncRNAs in the control of immaturity-stemness in the keratinocyte lineage. This hitherto poorly explored field will certainly have an impact on the domain of stem cells, notably in the skin model. Previously published profiling studies were focused on the search of lncRNAs involved in inflammatory skin diseases [32,33]. Moreover, in skin, the lncRNA *MALAT1* was shown to interact with the transcription factor MYC and bind to the promoter region of the Kinectin 1 (*KTN1*) gene, thus contributing to enhancement of epidermal growth factor (EGF) signaling in cutaneous squamous cell carcinoma [34]. Another lncRNA, *WAKMAR1*, has been documented as a regulator of keratinocyte migration, its deficiency impairing wound re-epithelialization, in the context of the diabetic foot [35]. Different approaches have been applied to attempt prediction of lncRNA biological functions. The approaches developed for this purpose include analyses of proximity with coding genes [6], machine learning co-expression analyses [33,36], and analyses based on sequence features such as k-mer profiles [37]. 

The computational approach developed in this study tended to link lncRNA signatures to functionally annotated pcRNA pathways. Co-expression correlation led us to propose prospective interacting maps depicted as regulation units or regulons centered on a particular lncRNA in potential interactions with a network of functionally associated pcRNAs (Figure 8). As a complementary approach, we also used Weighted Correlation Network Analysis (WGCNA) to depict dendrograms according to proximity matrixes (Figure 10). Interestingly, these analyses pointed to a selection of candidate lncRNAs potentially involved in the immaturity-stemness regulation processes in the keratinocyte lineage, available for dedicated comprehensive functional studies. Notably, transcripts related to the TGFB/BMP signaling network—including those of BMP, activin, membrane-bound inhibitor (*BAMBI*), Bone Morphogenetic Protein Receptor Type 1B (*BMPR1B*), DNA-binding protein inhibitor (*ID4*), and paired-like homeodomain transcription factor 2 (*PITX2*)— were associated with lncRNAs whose functions are not known. The same was true with transcripts linked to extracellular matrix biology, including those of collagen type X alpha 1 chain (*COL10A1*), matrix metallopeptidase 15 and 16 (*MMP15* and *MMP16*), and fibronectin (*FN1*). For example, the lncRNA referenced as *ENSG00000290931* that came out in our computational approach (gene localization on chr22 q11.1) is considered as a novel transcript. Another candidate lncRNA identified in our study, *MIR503HG*, has been previously described to be deregulated in tongue squamous cell carcinoma [38]. Of note, *MIR503HG* has been also associated with the promotion of pathophysiological processes mediated by dermal fibroblasts, including hypertrophic scar progression [39], and skin fibrosis, via interaction with TGFB/SMAD signaling [40].

Now, a major challenge in lncRNA biology is to define their specific biological functions. While we have a significant amount of experimental data and in silico methods for making function predictions based on the primary sequence for pcRNAs, we know little about the vast majority of lncRNAs, and no validated tool is available for performing sequence-based function predictions. Development of novel bio-informatics tools is thus greatly needed to enrich existing catalogs in functional annotation (for review, see [7]). In skin, lncRNA have retained attention as biomarkers and as modulators of biological pathways via miRNA sponging and regulation of pcRNA expression, in the context of cancer [41]. LncRNAs also retained attention in the contexts of skin ageing [42], wound healing [43], and inflammatory responses [44]. Concerning inflammatory skin diseases, a collection of RNA-seq datasets from in vitro and in vivo material was analyzed in silico through machine-learning methods, linking lncRNAs with cytokine signaling pathways [33].

Regarding future prospects for new candidates (lncRNA or other), the same strategy used for *KLF4* and *MXD4* studies will be implemented using functional genomics tools suitable for targeted known-down or over-expression, including shRNAs, small-interfering RNAs, antisense oligonucleotides, and over-expression systems. This experimental phase will be necessary to move from the stage of prospective identification to that of demonstrated biological functions, as we did for the KLF4 and *MXD4*/MAD4 transcription factors.

To conclude, concerning the primary topic of this study, which is deciphering the transcript network that regulates immaturity-stemness in human epidermal keratinocytes, the prospective computational approaches described here will give rise to the selection of candidates for classical experimental approaches, based on functional models.

## 4. Materials and Methods

Architecture of the computational approach is schematized in Appendix A. 

### 4.1. Library Pre-Processing and Read Quantitation

The RNA-seq transcriptome datasets corresponding to the *KLF4* and the *MXD4*/MAD4 studies model were obtained from the Gene Expression Omnibus (GEO) database under accession numbers GSE111786 and GSE202700, respectively [16,19]. Both *KLF4* and *MXD4* datasets were based on 3 biological replicates corresponding to the 2 cellular contexts: 3 wild type (WT) and 3 knock-down (KD) cellular contexts. A total of 17 RNA-seq libraries, including biological and technical replicates, were sequenced for the *KLF4* study, and 6 for the *MXD4* study Appendix A. The .sra files corresponding to each sample were downloaded using the prefetch tool v2.11.0 from the SRA Toolkit [45]. Subsequently, all .sra files were converted into .fastq files with the help of the fasterq-dump v2.11.0 from the SRA Toolkit. Samples were submitted to fastQC tool v.0.11.9 to perform quality control checks, before and after RNA-seq pre-processing [46]. The fastp tool v0.20.0 was used to remove Illumina adapters (Illumina, Inc., San Diego, CA, USA) and filter low-quality reads and bases [47]. Reads were aligned to the human genome (GRCh38 release 43) using STAR v2.7.10b [48]. Reads were quantified using the feature Counts tool v2.0.1 from the SubRead package [49]. Protein-coding transcripts (pcRNAs) long non-coding transcripts (lncRNAs) were identified using annotation from GENCODE GTF files.

### 4.2. Count Exploratory Analysis and Transcript Filtering

For each dataset, an expression matrix was built, filtering out the transcripts with no attributed reads in any sample. Then, all transcript detection corresponding to low signals were removed using the filterByExpr function (read counts per transcript ≤ 10 in at least 3 samples) from the edgeR package version v3.42.4 [50]. Technical variation was checked using dimension reduction analysis (principal component and multidimensional scaling analysis). In the absence of variation, technical replicates were combined by cumulating the counts. Batch effects were adjusted using the ComBat function from the R package sva version v3.36.0 [51], and a final expression matrix of 6 samples for each dataset was prepared in view of differential expression analysis. 

### 4.3. Differential Gene Expression Analysis

Differential gene expression analysis (DGE) was performed on each dataset, comparing the WT and KD cellular contexts using the edgeR package version v3.42.4. DGEList objects containing the raw expression matrix, data information, and gene annotation were first created for each dataset. Then, normalization factors were calculated using the calcNormFactors function and the trimmed mean of M-values (TMM) method, in order to scale the library sizes. The mean–variance relationship was then estimated using the estimateDisp function. Moderated *p*-values were calculated by fitting a quasi-likelihood negative binomial generalized log-linear model to the count data, using the empirical Bayes method (glmQLFit function). Transcripts that satisfied the following criteria were tagged as differentially expressed (DE): absolute fold change (|fold-change|) > 1.5 and False Discovery rate (FDR) < 0.05. Visualization plots were constructed with the ggplot2 and pheatmap packages, on the R platform.

### 4.4. Functional Enrichment Analysis and Gene Set Enrichment Analysis of pcRNAs

To investigate the biological significance of pcRNAs detected as differentially expressed (DE) in KD versus WT cells, in the *KLF4* and the *MXD4* transcriptome datasets, both Over-Representation Analysis (ORA) and Gene Set Enrichment Analysis (GSEA) were conducted. ORA is a statistical method used to determine whether transcripts from a priori defined set are present more than one would be expected (over-represented) in a particular subset of transcripts of interest [52]. The hypergeometric tests were applied using the function enricher from the clusterProfiler package v4.8.2 from the R platform. Seven different gene set libraries downloaded from from Enrichr database [53] were used for ORA: Kyoto Encyclopedia of Genes and Genomes (KEGG), Reactome pathways, Gene ontology biological processes (GOBP), Bioplanet, Hallmarks of genes, ChIP Enrichment Analysis 2013 (ChEA), and Tabula Muris. Enrichment *p*-values < 0.05 were considered as statistically significant. Graphical representation was performed using the ggplot package from the R platform. GSEA is a rank-based computational approach that determines whether pre-defined groups of transcripts show statistically significant and concordant differences between two biological states [54]. Analysis and graphical representation were performed using the GSEA desktop application v4.1.0. This unsupervised GSEA algorithm was applied to the study of the entire normalized expression signals of pcRNAs from each RNA-seq dataset. Pre-selected transcript sets were from the Hallmarks of genes and MiSigDB databases.

### 4.5. Functional Enrichment Analysis of DE Transcripts Common to the KLF4 and MXD4 Datasets

A Venn diagram analysis was applied to the complete list of DE transcripts from the *KLF4* and *MXD4* datasets, using the Venndetail package from the R platform. In order to compare the effect of the Log_2_ Fold change and visualize the sense of the regulation (up or down) on DE transcripts identified in common in the *KLF4* and *MXD4* datasets, we built scatter plots for pcRNAs and for lncRNAs, using the ggplot package from R. Concerning the signature of commonly identified DE pcRNAs, ORA was applied to the *KLF4* and *MXD4* datasets for functional interpretation, using the compareCluster function with the option enricher from the clusterProfiler package. Up-modulated and down-modulated pcRNAs identified in both sets were distinguished. A manual classification of enriched terms selected for their relevance for the topic of the study was made for functional interpretation. 

### 4.6. Prospective Building of lncRNA-pcRNA Transcriptional Regulatory Networks (TRNs)

A first approach implemented for interpretation of lncRNA–pcRNA signatures consisted in the prospective building of Transcriptional Regulatory Networks (TRNs), using the RTN v.2.24.0 R package [55]. The RTN package tests the association of given elements and potential targets, based on expression modulations. Here, the tested elements were the DE lncRNAs identified in common from the *KLF4* and the *MXD4* models, and the potential targets were the common DE pcRNAs. The TRN approach comprised two main steps. The first step was Transcriptional Networks Inference (TNI) that results in the definition of regulatory units called regulons. The TNI analysis included the computation of Mutual Information (MI) between particular candidate regulators and the potential targets, removing non-significant associations by permutation analysis (n = 1000 permutations). The removal of unstable interactions is achieved by bootstrapping and application of the reconstruction of Accurate Cellular Networks (ARACNe) method. The second step was Transcriptional Network Analysis (TNA). Regulons that comprised at least 15 interactions or more were tested for expression association with the pcRNA signature by GSEA. The resulting networks were visualized using the Cytoscape software v3.7.2. Venn diagram analysis was applied to integrate the results from both datasets.

### 4.7. Prospective Definition of lncRNA–pcRNA Interactions by Consensus Weighted Gene Co-Expression Network Analysis (WGCNA)

As a second approach implemented for interpretation of lncRNA–pcRNA signatures consisted in Consensus Weighted Gene Co-expression Network analysis (WGCNA), which aims at identifying significant modules and hub genes associated with particular phenotypes, using correlation analysis among transcripts across RNA-seq samples. Consensus co-expression network analysis differs from the standard co-expression network analysis workflow by constructing individual networks across distinct datasets, and then computing an integrated co-expression network. The consensus WGCNA was computed using the entire expression datasets from the *KLF4* and *MXD4* studies, following the step-by-step network construction and module detection of consensus network analysis from WCGNA package [56]. The soft threshold was set at 8 for the adjacency calculation, since it is the lowest value for which the scale-free topology index reaches 0.90 and the connectivity measures decrease for both datasets (Appendix A). Then, modules correlated with the two KD contexts were considered for network visualization. Graphical network visualization was performed using the RedeR v2.4.3 and the TreeAndLeaf v1.12.0 packages from the R platform [57,58].

### 4.8. Prospective Search of pcRNA–lncRNA Interactions Using Databases of miRNAs

A list of common lncRNAs identified by TRN and WGCNA analysis was submitted to the StarBase ENCORI tool [59]. A resulting set of predicted lncRNA associated-miRNA was obtained using the miRNA–lncRNA interaction networks obtained from this encyclopedia of RNA interactomes. A threshold of target-directed microRNA (miRNA) degradation (TDMD) score > 0.8 was applied. The global set of miRNAs identified for each lncRNA was submitted to miRTargetLink 2.0 with a “strong validated” edit network (validated by experimentation such as luciferase reporter assays, qRT-PCR, Western Blot, etc.). The resulting networks were visualized using the Cytoscape software v3.7.2. 

## Figures and Tables

**Figure 1 ijms-25-03353-f001:**
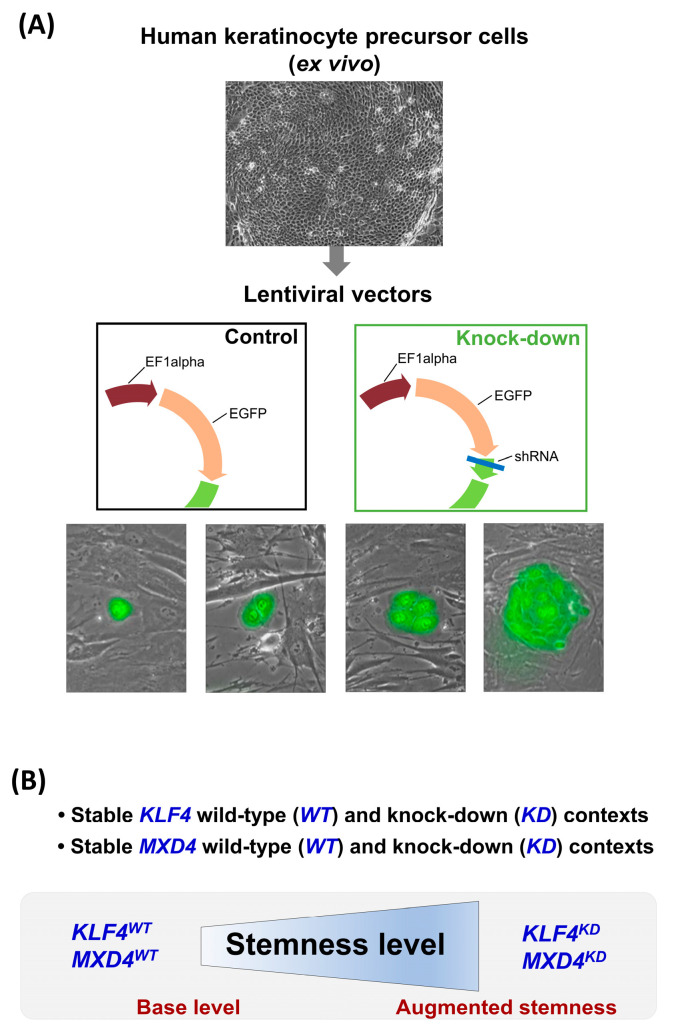
Schematic summary of the cellular model. (**A**) Keratinocyte precursor cells corresponding to the progeny of a holoclone (i.e., keratinocyte stem cell) were transduced with a control lentiviral vector driving GFP expression (control = wild-type) or with a vector designed for targeted gene knock-down (driving expression of GFP and of a specific shRNA). Transduced cells were sorted by flow cytometry based on GFP fluorescence. (**B**) Wild-type (WT) and knock-down (KD) contexts were generated for two candidate transcription factors: *KLF4* and *MXD4*/MAD4. The two knock-down contexts were documented to promote an augmented stemness cellular status (see the source publications [16,19]).

**Figure 2 ijms-25-03353-f002:**
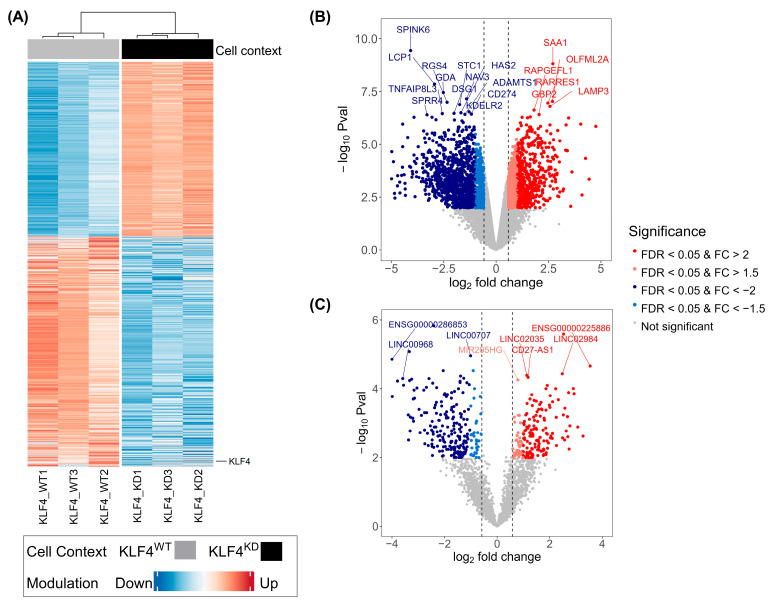
Protein-coding RNA (pcRNA) and lncRNA transcripts identified as differentially expressed (DE) between *KLF4*^KD^ and *KLF4*^WT^ cells by RNA-Seq. (**A**) Heatmap of mean expression levels [log_2_ (CPM)]) of 2712 lncRNA and protein-coding transcripts found DE. Blue are down-modulated and red are up-modulated transcripts. The *KLF4* expression is depicted on the right side of the heatmap. (**B**,**C**) Volcano plot representation of log_2_ fold change and −log_10_ *p* values resulting from differential expression analysis. Statistically significant (FDR < 0.05) up-modulated transcripts with Fold change (FC) > 2 are red dots and 1.5 < FC < 2 are orange dots. Statistically significant (FDR < 0.05) down-modulated transcripts with FC < −2 are blue dots and −1.5 > FC > −2 are light blue dots. The top differentially expressed transcripts are labeled. (**B**) A volcano plot of pcRNA transcripts analysis. (**C**) A volcano plot of lncRNAs analysis. Expression modulation is classified following the expression level in the KD condition. Differential expression follows FDR < 0.05 and absolute FC > 1.5 (EdgeR package).

**Figure 3 ijms-25-03353-f003:**
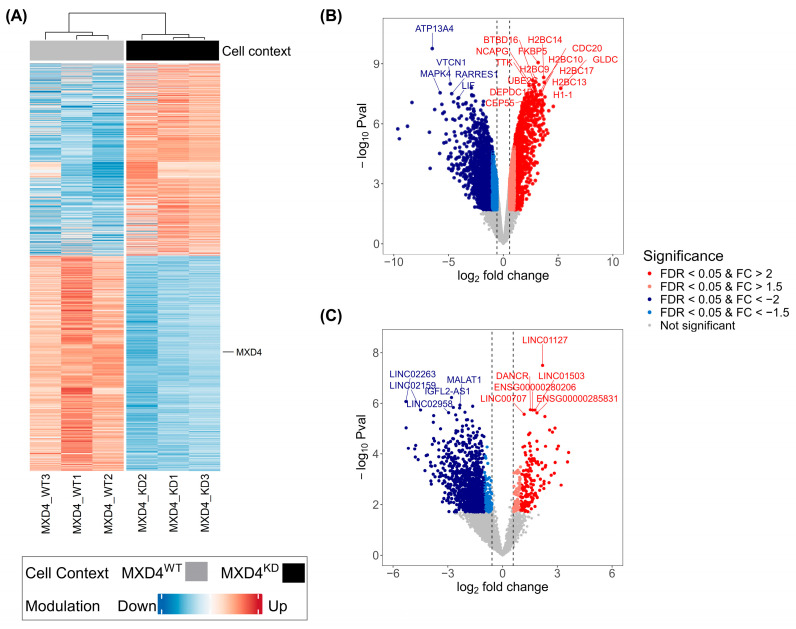
Protein-coding RNA (pcRNA) and lncRNA transcripts identified as differentially expressed (DE) between *MXD4*^KD^ and *MXD4*^WT^ cells by RNA-Seq. (**A**) Heatmap of mean expression levels [log_2_ (CPM)] of 6664 lncRNA and protein-coding transcripts found DE. Blue are down-modulated and red are up-modulated transcripts. *MXD4* expression is depicted on the right side of the heatmap (**B**,**C**) Volcano plot representation of log_2_FC and −log_10_ *p* values resulting from differential expression analysis. Statistically significant (FDR < 0.05) up-modulated transcripts with FC > 2 are red dots and FC between 1.5 and 2 are orange dots. Statistically significant (FDR < 0.05) down-modulated transcripts with FC < −2 are blue dots and FC between −1.5 and −2 are light blue dots. The top differentially expressed transcripts are labeled. (**B**) A volcano plot of pcRNA transcripts analysis. (**C**) A volcano plot of lncRNAs analysis. Expression modulation is classified following the expression level in the KD condition. Differential expression follows FDR < 0.05 and absolute FC > 1.5 (EdgeR package).

**Figure 4 ijms-25-03353-f004:**
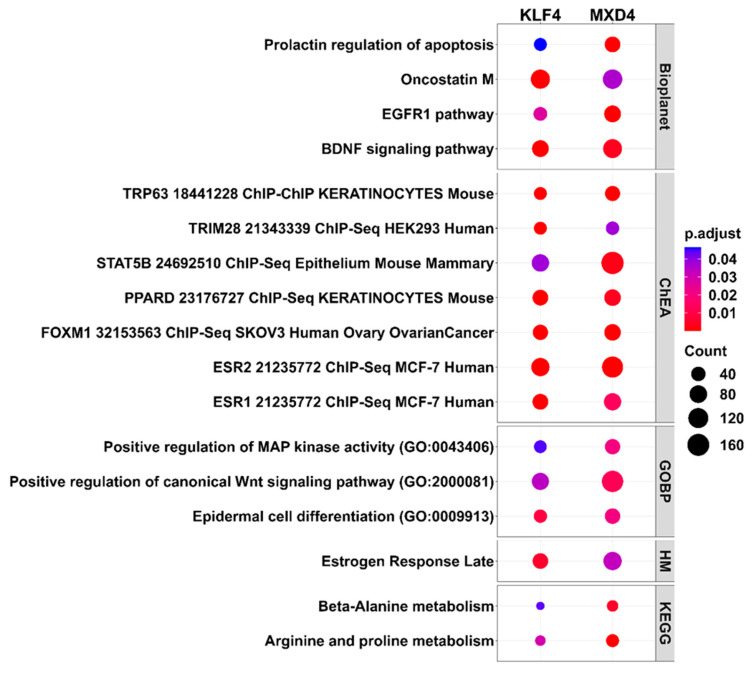
Functional enrichment analysis of differentially expressed transcripts in the *KLF4* and *MXD4* datasets using Over-Representation Analysis (ORA). The dot plot displays the *p*-adjusted values (*p*.adjust—dot color ramp) of 17 pathway terms found enriched in DE pcRNA transcripts identified in the *KLF4* and *MXD4* datasets. Seven pathway gene sets were downloaded from the Enrichr database and used for ORA with ClusterProfiler package on R. Kyoto Encyclopedia of Genes and Genomes (KEGG), Reactome pathways, Gene Ontology Biological Processes (GOBP), Bioplanet, MiSigDB Hallmarks of genes (HM), ChIP Enrichment Analysis 2013 (ChEA) databases were analyzed. Pathway terms were considered enriched when *p* adjusted values < 0.05. Dot size represents the number of enriched transcripts for each pathway (Count).

**Figure 5 ijms-25-03353-f005:**
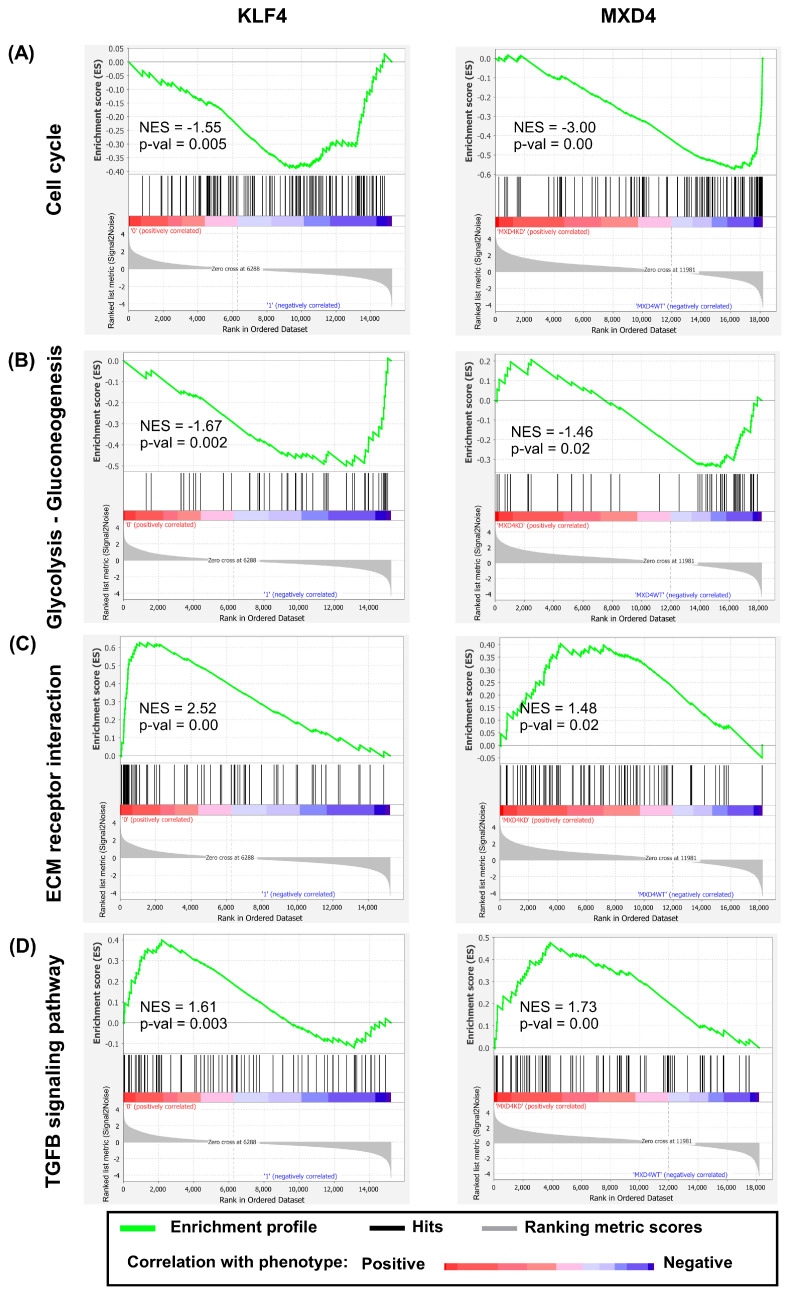
KEGG-based functional exploration by Gene Set Enrichment Analysis (GSEA). Enrichment plots of four KEGG pathways found enriched in DE transcripts identified both in the *KLF4* (**left** panel) and the *MXD4* model (**right** panel). (**A**) Cell cycle, (**B**) glycolysis–gluconeogenesis, (**C**) extracellular matrix (ECM) receptor interaction, and (**D**) TGFB signaling pathway. In each graph, the red left part represents transcripts correlated with the KD context, and the blue right part represents transcripts correlated with the WT context. The vertical black lines indicate the position of each of the transcripts on the ranked list. The green curve denotes the ES (enrichment score) curve. Nominal *p* values and normalized enrichment scores (NES) are shown.

**Figure 6 ijms-25-03353-f006:**
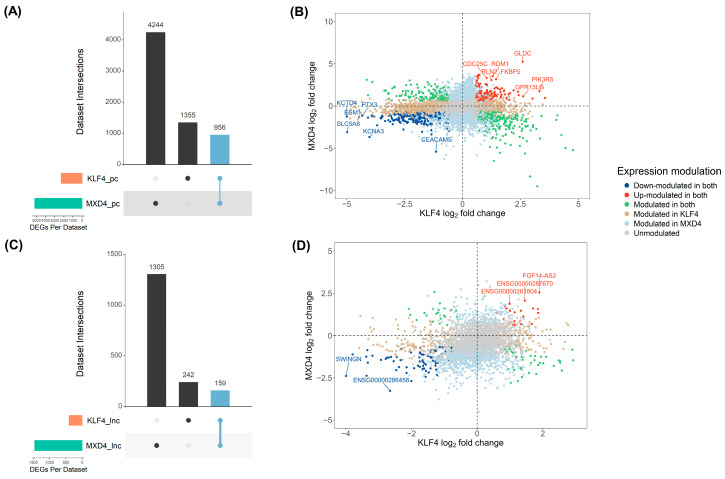
Shared DE transcripts identified both in the *KLF4* and the *MXD4* datasets. Individual differential expression analyses of the *KLF4* and *MXD4* datasets were integrated. Upset plots display the intersection of DE transcripts of (**A**) pcRNAs and (**C**) lncRNAs. Shared DE transcripts are visualized as light blue bars. Scatter plots are the representation of log_2_FC results of *KLF4* dataset analysis (x-axis) and *MXD4* dataset analysis (y-axis) for pcRNA (**B**) and lncRNA (**D**) transcripts. Transcripts found down- or up-modulated in both models are visualized as blue and red dots, respectively. Transcripts with opposite modulation in *KLF4^KD^* and *MXD4^KD^* cells (down- versus up- or up- versus down-modulated) are visualized as green dots. DE transcripts identified only in the *KLF4* model are visualized as tan dots and DE transcripts identified only in *MXD4* model are visualized as light blue dots.

**Figure 7 ijms-25-03353-f007:**
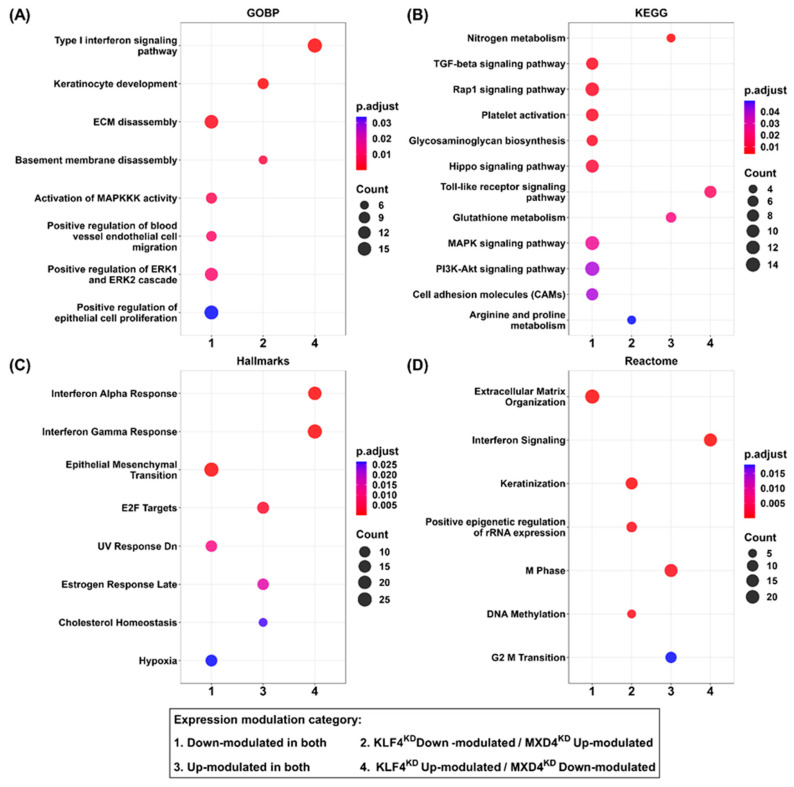
Over-Representation Analysis (ORA) of common modulated transcripts in the *KLF4* and *MXD4* models. Dot plots for (**A**) Gene Ontology Biological Processes (GOBP), (**B**) Kyoto Encyclopedia of Genes and Genomes (KEGG), (**C**) MiSigDB Hallmarks of genes, and (**D**) Reactome pathways terms found enriched on pcRNA transcripts modulated both in the *KLF4^KD^* and the *MXD4^KD^* contexts. Pathway terms were considered enriched when *p* adjusted values < 0.05. Dot size is proportional to the number of enriched transcripts for each pathway (Count). Dot color ramp represents the adjusted *p*-value (*p*.adjust).

**Figure 8 ijms-25-03353-f008:**
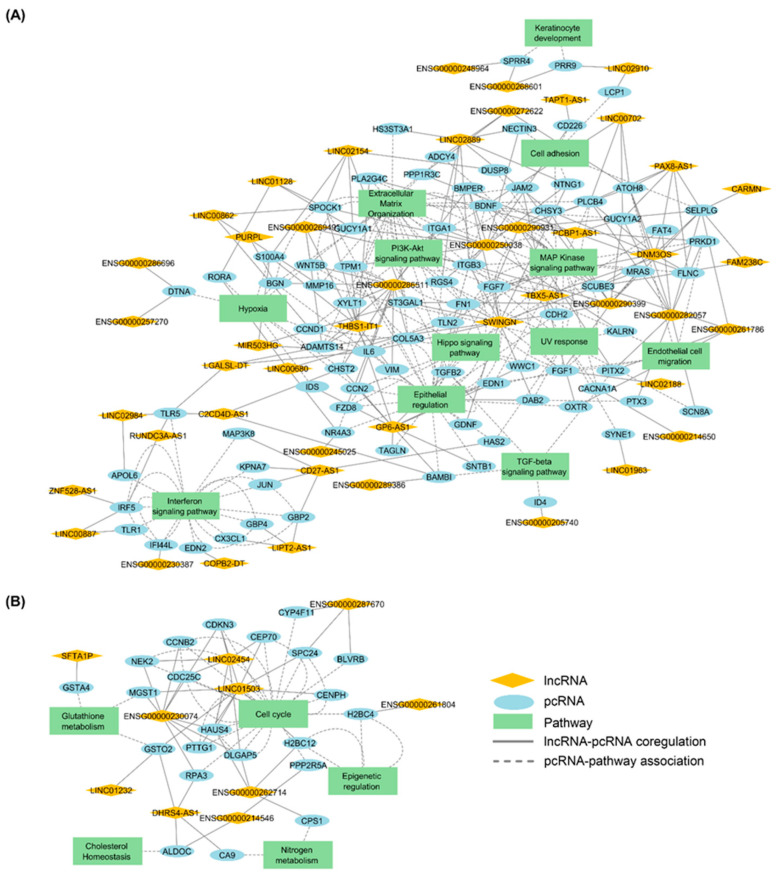
Transcriptional Regulatory Networks (TRNs) based on DE pcRNAs common to *KLF4^KD^* and *MXD4^KD^* cells, in association with lncRNAs. (**A**) Coregulatory network of 47 regulons (47 lncRNAs associated to 82 pcRNAs) and their corresponding enriched pathways terms. (**B**) Coregulatory network of 10 regulons (10 lncRNAs associated to 22 pcRNAs) and their enriched pathways terms.

**Figure 9 ijms-25-03353-f009:**
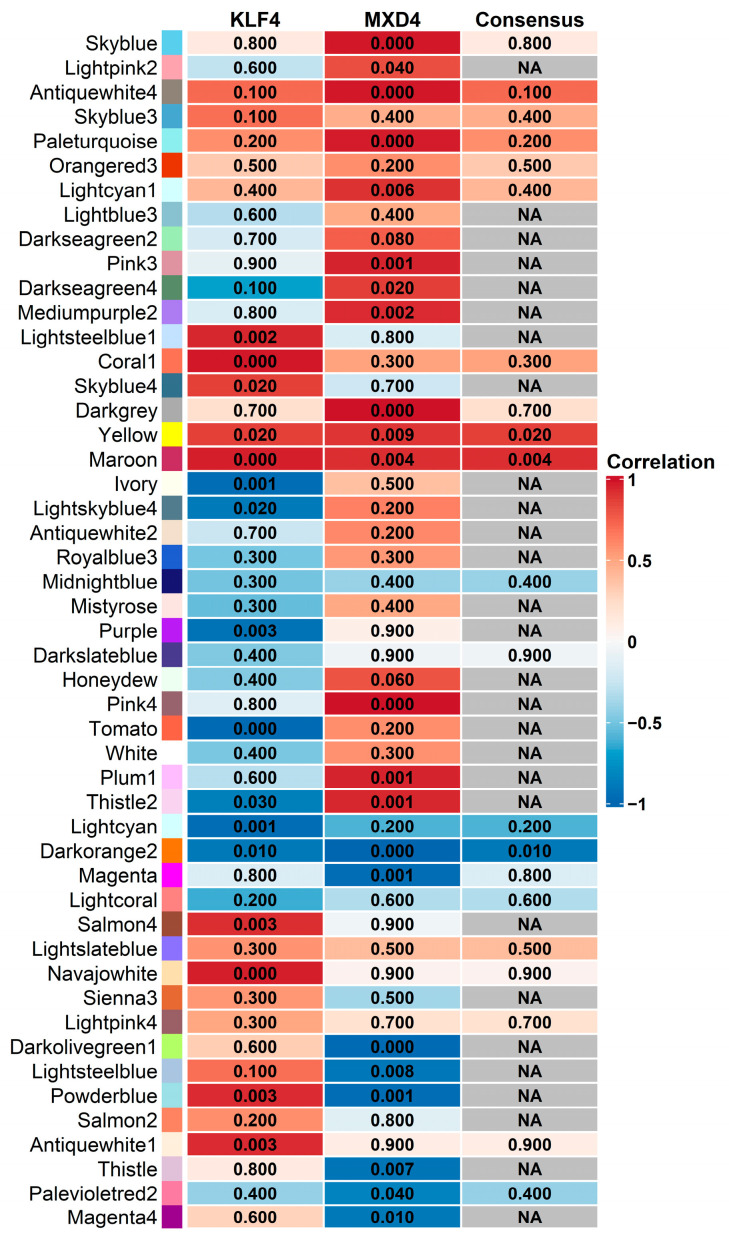
Relationships of consensus module genes with DE transcripts from the *KLF4* and *MXD4* datasets. Each row in the heatmap corresponds to a module, and each column to each cell context and the network consensus. Numbers in the heatmap report the correlation *p* values of the corresponding module and the cellular context, and the heatmap is color-coded by correlation values according to the color legend. NA indicates not applicable consensus between the two models.

**Figure 10 ijms-25-03353-f010:**
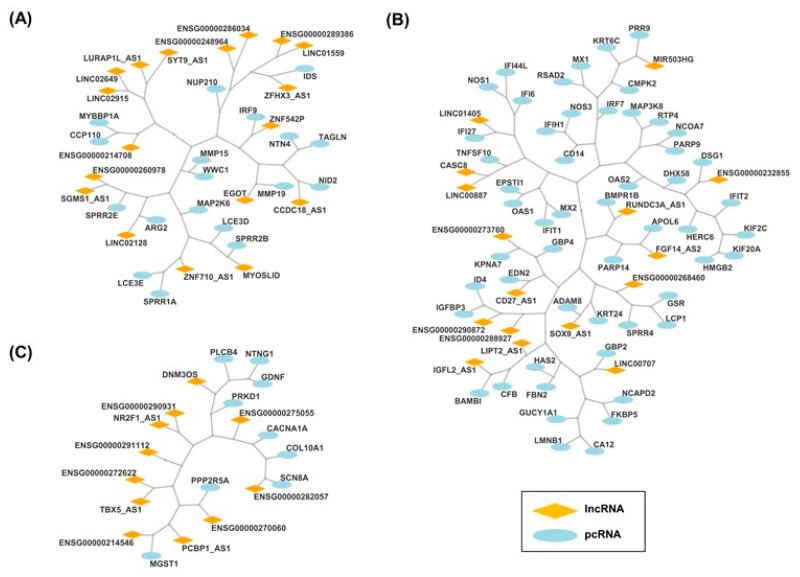
Transcript modules common to the *KLF4* and *MXD4* cellular models, defined by consensus WGCNA. Tree-and-leaf representation of the consensus modules identified in common from the *KLF4* and *MXD4* models. (**A**) Dark orange, (**B**) maroon, and (**C**) yellow modules. Diamond and ellipse nodes represent lncRNAs and protein-coding RNAs (pcRNAs), respectively. Branch ramifications represent the clustered nodes and distances between nodes (edges) represent the calculated distance from consensus correlation values obtained with WGCNA.

**Figure 11 ijms-25-03353-f011:**
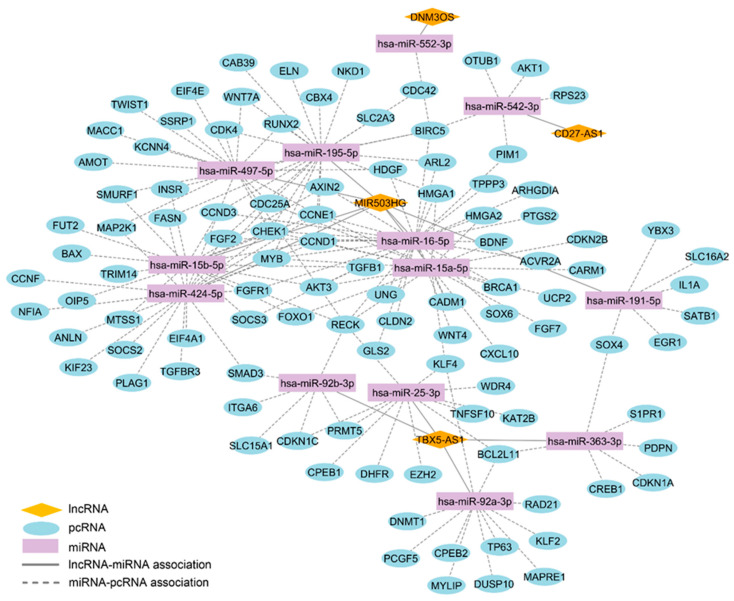
The lncRNA-associated competitive endogenous RNA (ceRNA) network. Illustration of the in silico constructed lncRNA/miRNA/mRNA ceRNA network. The diamonds represent lncRNAs, the ellipses represent pcRNAs, and the rectangles represent miRNAs. LncRNA–miRNA interactions are solid gray lines and miRNA–pcRNA interactions are dashed gray lines.

**Figure 12 ijms-25-03353-f012:**
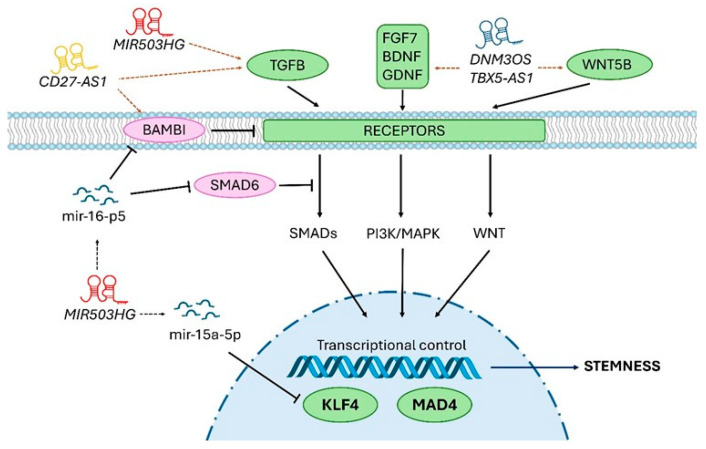
LncRNAs potentially involved in the immaturity-stemness regulation processes in the human keratinocyte lineage. Schematic representation of the hypothetical lncRNAs participation on TGFB, PI3K/MAPK, and WNT pathways. Activator and repressor pathway proteins are green and purple, respectively; lncRNAs are loop RNA structures; and involved miRNAs are small dark blue structures. Confirmed molecule interactions are black arrows, co-expression interactions are orange arrows.

**Table 1 ijms-25-03353-t001:** Numbers of differentially expressed transcripts in *KLF4^WT^* and *KLF4^KD^* keratinocytes.

Biotype	Modulation in *KLF4^KD^* versus *KLF4^WT^* Cells	Number of Transcripts *	
		Absolute Fold-Change (FC) > 2	Absolute Fold-Change 2 > (FC) > 1.5
pcRNAs	Up-modulation	490	505
	Down-modulation	939	377
lncRNAs	Up-modulation	136	39
	Down-modulation	193	33

* False discovery rate (FDR) < 0.05.

**Table 2 ijms-25-03353-t002:** Top 20 most significant DE pcRNAs in *KLF4^WT^* versus *KLF4^KD^* keratinocytes sorted according to increasing FDR values.

Gene Symbol	Log_2_FC	Log_2_ CPM	FDR	Absolute FC
SPINK6	−4.08	4.23	6.6 × 10^−6^	16.87
SAA1	2.71	6.53	1.4 × 10^−5^	6.53
CLIC5	−5.23	2.26	2.6 × 10^−5^	37.57
LCP1	−2.94	4.20	6.3 × 10^−5^	7.68
GDA	−2.52	3.26	1.3 × 10^−4^	5.74
NAV3	−1.41	5.25	2.1 × 10^−4^	2.67
OLFML2A	2.69	3.48	2.1 × 10^−4^	6.47
RGS4	−2.34	4.37	2.1 × 10^−4^	5.05
RARRES1	2.47	4.34	2.1 × 10^−4^	5.55
STC1	−1.74	6.42	2.3 × 10^−4^	3.33
LAMP3	2.57	3.23	2.6 × 10^−4^	5.94
RAPGEFL1	1.81	5.74	3.6 × 10^−4^	3.51
KDELR2	−1.29	7.62	3.6 × 10^−4^	2.44
HAS2	−1.68	6.14	3.6 × 10^−4^	3.21
DSG1	−2.01	6.74	3.6 × 10^−4^	4.03
TNFAIP8L3	−2.57	2.29	3.6 × 10^−4^	5.95
CD274	−1.18	5.45	3.6 × 10^−4^	2.27
ADAMTS1	−1.49	7.50	3.6 × 10^−4^	2.82
GBP2	2.05	3.69	3.6 × 10^−4^	4.13
SPRR4	−3.30	1.44	3.6 × 10^−4^	9.86

**Table 3 ijms-25-03353-t003:** Top 20 most significant DE lncRNAs in *KLF4^WT^* versus *KLF4^KD^* keratinocytes sorted according to increasing FDR values.

Gene Symbol	Log_2_FC	Log_2_ CPM	FDR	Absolute FC
LNCOG	−2.41	2.01	0.00057	5.31
LINC02188	2.53	1.49	0.00067	5.78
ENSG00000250038	−3.34	−0.22	0.00110	10.16
LINC00707	−1.01	5.69	0.001	2.02
ENSG00000286853	−4.00	−1.04	0.001	16.04
ENSG00000225886	3.54	−1.14	0.002	11.62
ENSG00000255050	−2.05	0.66	0.002	4.15
DIRC3	−4.03	−0.97	0.002	16.39
LINC02984	2.48	0.70	0.002	5.59
LINC02035	1.13	3.32	0.002	2.20
WAKMAR2	−1.39	2.77	0.002	2.63
CD27-AS1	1.18	2.68	0.003	2.26
LINC01559	−1.27	2.21	0.003	2.41
HAGLR	−2.59	−0.08	0.003	6.04
LINC00968	−3.59	−0.12	0.003	12.04
HOXB-AS3	−4.33	−0.69	0.003	20.10
ENSG00000287963	−3.25	−1.29	0.003	9.54
TBX5-AS1	−3.33	0.69	0.003	10.06
ENSG00000224888	−3.79	−1.81	0.003	13.85
APCDD1L-DT	−1.41	2.04	0.003	2.66

**Table 4 ijms-25-03353-t004:** Numbers of differentially expressed transcripts in *MXD4^WT^* and *MXD4^KD^* keratinocytes.

Biotype	Modulation in *MXD4^KD^* versus *MXD4^WT^* Cells	Number of Transcripts *	
		Absolute Fold-Change (FC) > 2	Absolute Fold-Change 2 > (FC) > 1.5
pcRNAs	Up-modulation	1218	1713
	Down-modulation	1554	715
lncRNAs	Up-modulation	151	69
	Down-modulation	1122	122

* False discovery rate (FDR) < 0.05.

**Table 5 ijms-25-03353-t005:** Top 20 most significant DE pcRNAs in *MXD4^WT^* versus *MXD4^KD^* keratinocytes.

Gene Symbol	Log_2_FC	Log_2_ CPM	FDR	Absolute FC
ATP13A4	−6.49	3.02	3.65 × 10^−6^	89.86
FKBP5	3.18	5.48	9.27 × 10^−6^	9.04
H2BC14	3.68	4.92	2.29 × 10^−5^	12.85
TRIP13	2.66	5.13	2.29 × 10^−5^	6.32
H2BC7	2.91	5.02	2.29 × 10^−5^	7.53
H2BC10	3.73	5.36	2.29 × 10^−5^	13.31
VTCN1	−4.84	2.68	2.29 × 10^−5^	28.73
H2BC15	2.71	4.56	2.29 × 10^−5^	6.53
H2BC9	3.25	5.68	2.29 × 10^−5^	9.54
AURKA	2.35	5.50	2.29 × 10^−5^	5.12
H2BC6	2.82	4.12	2.29 × 10^−5^	7.04
MELTF	−2.91	5.63	2.29 × 10^−5^	7.52
SKA1	2.84	3.50	2.29 × 10^−5^	7.16
GLDC	5.24	4.57	2.29 × 10^−5^	37.91
CPVL	2.91	2.95	2.29 × 10^−5^	7.49
CDC20	3.39	6.43	2.29 × 10^−5^	10.45
SKA3	2.54	3.40	2.29 × 10^−5^	5.82
FAM83D	2.65	4.62	2.29 × 10^−5^	6.29
MAPK4	−5.77	1.31	2.29 × 10^−5^	54.60
MCM10	2.22	4.42	2.29 × 10^−5^	4.65

**Table 6 ijms-25-03353-t006:** Top 20 most significant DE lncRNAs in *MXD4^WT^* versus *MXD4^KD^* keratinocytes.

Gene Symbol	Log_2_FC	Log_2_ CPM	FDR	Absolute FC
LINC01127	2.19	3.70	0.00002	4.57
IGFL2-AS1	−2.80	3.41	0.00006	6.96
LINC02263	−5.30	−0.07	0.00008	39.33
ENSG00000260978	−2.34	1.08	0.00009	5.05
TMEM51-AS1	−1.64	2.83	0.00010	3.11
ENSG00000268460	−2.70	1.21	0.00010	6.49
MALAT1	−2.36	7.70	0.00010	5.12
DANCR	1.52	4.90	0.00011	2.87
LINC02159	−4.49	−0.22	0.00012	22.53
LINC01503	1.77	4.02	0.00012	3.42
ENSG00000280206	1.65	2.93	0.00012	3.13
ENSG00000261324	−2.06	2.64	0.00013	4.17
LINC02958	−2.96	1.43	0.00014	7.77
ENSG00000285831	1.88	2.30	0.00014	3.69
LINC00707	1.18	4.86	0.00015	2.27
ENSG00000259354	−2.57	0.40	0.00015	5.94
LINC02593	−3.78	−0.21	0.00016	13.69
ENSG00000266088	2.32	2.34	0.00017	4.99
ENSG00000205890	−1.92	2.11	0.00020	3.77
ELF3-AS1	−2.50	2.34	0.00020	5.67

**Table 7 ijms-25-03353-t007:** Signature of 14 common modulated lncRNAs in the *KLF4* and *MXD4* models, identified as coregulatory elements by WGCNA and TNA methods.

Gene Symbol	*KLF4*		*MXD4*		Expression Modulation
	Log_2_FC	FDR	Log_2_FC	FDR	
CD27-AS1	1.18	0.003	−0.59	0.031	Up in *KLF4*/Down in *MXD4*
PCBP1-AS1	−0.78	0.032	−0.69	0.009	Down in both
LINC00887	2.93	0.004	−1.47	0.013	Up in *KLF4*/Down in *MXD4*
ENSG00000214546	1.33	0.046	1.46	0.016	Up in both
MIR503HG	−2.45	0.003	−1.27	0.000	Down in both
DNM3OS	−2.69	0.003	−1.31	0.020	Down in both
ENSG00000248964	−1.54	0.006	1.18	0.005	Down in *KLF4*/Up in *MXD4*
LIPT2-AS1	1.09	0.016	−1.03	0.033	Up in *KLF4*/Down in *MXD4*
TBX5-AS1	−3.33	0.003	−1.61	0.007	Down in both
RUNDC3A-AS1	1.04	0.011	−0.95	0.009	Up in *KLF4*/Down in *MXD4*
ENSG00000272622	−1.76	0.026	−1.30	0.023	Down in both
ENSG00000282057	−2.15	0.009	−2.21	0.008	Down in both
ENSG00000289386	1.58	0.030	−2.21	0.004	Up in *KLF4*/Down in *MXD4*
ENSG00000290931	−3.38	0.009	−1.25	0.041	Down in both

## Data Availability

RNA-seq transcriptome datasets are available in the Gene Expression Omnibus (GEO) database under accession numbers GSE111786 and GSE202700.

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
