# Peer review of "Deciphering “Immaturity-Stemness” in Human Epidermal Stem Cells at the Levels of Protein-Coding and Non-Coding Genomes: A Prospective Computational Approach"

_ijms, 2024, doi:10.3390/ijms25063353_

Round 1
Reviewer 1 Report
Comments and Suggestions for Authors
In this paper, the authors have studied the KLF4/TGFB1 and MAD4/MAX/MYC signaling pathways in controlling the immaturity-stemness versus differentiation fate of keratinocyte stem and precursor cells from human interfollicular epidermis. They found that down-modulation of these pathways promoted an augmented stemness cellular status, accompanied by significant transcriptional changes. A computational approach was used to integrate the coding and non-coding genomes, with a focus on long non-coding RNAs. Moreover, data of micro-RNAs (miRNAs) with validated functions were also interrogated to identify miRNAs that could link protein-coding and non-coding transcripts. Eventually, by integrating all these data together, the authors defined putative regulons, containing original pro-stemness candidate effectors for functional validation approaches. This is a very nice paper that provides a prospective picture of the complex constellation of transcripts regulating keratinocyte stemness status. I recommend publishing this work in IJMS journal.
Author Response
Manuscript ID: ijms-2887060
Point-by-point responses to Reviewer’s comments and suggestions:
Black font: reviewer’s comments and suggestions.
Blue font: author’s responses.
Red font: addition in the manuscript.
Review Report 1
Comments and Suggestions for Authors
In this paper, the authors have studied the KLF4/TGFB1 and MAD4/MAX/MYC signaling pathways in controlling the immaturity-stemness versus differentiation fate of keratinocyte stem and precursor cells from human interfollicular epidermis. They found that down-modulation of these pathways promoted an augmented stemness cellular status, accompanied by significant transcriptional changes. A computational approach was used to integrate the coding and non-coding genomes, with a focus on long non-coding RNAs. Moreover, data of micro-RNAs (miRNAs) with validated functions were also interrogated to identify miRNAs that could link protein-coding and non-coding transcripts. Eventually, by integrating all these data together, the authors defined putative regulons, containing original pro-stemness candidate effectors for functional validation approaches. This is a very nice paper that provides a prospective picture of the complex constellation of transcripts regulating keratinocyte stemness status. I recommend publishing this work in IJMS journal.
Response: The authors would like to thank reviewer 1 for the very positive evaluation of our manuscript.

Reviewer 2 Report
Comments and Suggestions for Authors
Here the authors describe their computational approach to study gene expression related to keratinocyte 'stemness'. They used differential expression data from knockdown keratinocyte models and analysed profiles of protein-coding and long non-coding RNAs. This is interesting as keratinocyte amplification versus differentiation is critical, and current signatures, especially in respect of non-coding RNAs, are insufficient.
- The findings are clear and clearly described. Obviously, while some approaches are innovative, they should be discussed with care and critical points and hypothetical results should be more clearly pointed out and evaluated. The approach is more straight forward regarding protein-coding RNAs; the mentioned "conventional genome" is not very clear though and it should be clarified what's known and what is new, with cautious validation.
- Critically, the results depend on sources of suitable tissues, i.e. the "adequate cellular model". This has to be more clearly defined. An "adequate cellular model to decipher the stemness in a human tissue" is a very broad characteristic; it is unclear if this model is really suitable for such a generalised objective. Limitations should be pointed out, and even the justification with regard to the skin only refers to few - though important - examples of clinical applications.
- More difficult, the cellular expression models have to be justified. It is well known that specific knockdown approaches can have broad effects on the cellular homeostasis. It is briefly stated that the results obtained here would demonstrate that the model was suitable for the network analysis; however, a more critical review taking the predictive nature of results into account would be needed.
- An innovative part is the profiling of lncRNAs alongside protein-coding RNA signatures. These data identify new strategies for experimental studies and require such validations. It should be clarified to which extent they can be relevant: Linking lncRNA signatures to functionally annotated protein-coding RNA pathways seems to be promising; given that the protein-coding RNA networks have been developed from an experimental 'stemness' model, the validity of this alignment should be more critically discussed.
Minor point:
- The first paragraph of the Results section (Cellular models) refers to previous publications of the authors. Their generation is not done and not described here and could be more suitably arranged in the text.
Author Response
Manuscript ID: ijms-2887060
Point-by-point responses to Reviewer’s comments and suggestions:
Black font: reviewer’s comments and suggestions.
Blue font: author’s responses.
Red font: addition in the manuscript.
Review Report 2
Comments and Suggestions for Authors
Here the authors describe their computational approach to study gene expression related to keratinocyte 'stemness'. They used differential expression data from knockdown keratinocyte models and analysed profiles of protein-coding and long non-coding RNAs. This is interesting as keratinocyte amplification versus differentiation is critical, and current signatures, especially in respect of non-coding RNAs, are insufficient.
- The findings are clear and clearly described. Obviously, while some approaches are innovative, they should be discussed with care and critical points and hypothetical results should be more clearly pointed out and evaluated. The approach is more straight forward regarding protein-coding RNAs; the mentioned "conventional genome" is not very clear though and it should be clarified what's known and what is new, with cautious validation.
Response: We thank and appreciate questions and valuable suggestions from reviewer 2.We have tried to consider them wherever possible. We are completely aware that computational approaches are only prospective, and constitute a basis for the identification of candidates, which must then be studied using functional experimental approaches.
We agree that the term ‘conventional genome’ was unclear. As our objective was to designate the genome responsible for coding the proteome, we substituted it with the term 'protein-coding genome' (page 1, line 43).
- Critically, the results depend on sources of suitable tissues, i.e. the "adequate cellular model". This has to be more clearly defined. An "adequate cellular model to decipher the stemness in a human tissue" is a very broad characteristic; it is unclear if this model is really suitable for such a generalised objective. Limitations should be pointed out, and even the justification with regard to the skin only refers to few - though important - examples of clinical applications.
Response: In our previous studies, in order to ensure the absence of unwanted effects due to shRNA-based technology, an in-depth characterization of the functional and molecular properties of the knockdown cells was carried out. The functional gain associated with increased immaturity-stemness was established on the basis of reference tests and controls, including clonogenicity tests, long-term growth, and 3D epidermal regeneration, coupled with analyzes of the genome integrity. In addition, the relevance of the holoclone model as a representative system of epithelial stem cells is established on the basis of reference publications and clinical achievements, cited in the manuscript:
- Barrandon et al, 1987, doi: 10.1073/pnas.84.8.230
- Droz‐Georget Lathion et al., 2015, doi: 10.15252/emmm.201404353
- Hirsch et al., 2017, doi: 10.1038/nature24487
- Kueckelhaus et al., 2021, doi: 10.1056/NEJMoa2108544
We have added the following sentence to more clearly define de notion of "adequate cellular model", as follows (page 2, lines 64-66). In addition, “adequate” was changed into “relevant” (page 2, line 49)
In summary, the cellular model of holoclone keratinocytes has the advantage of connecting basic researches aimed at acquiring knowledge on epidermal stem cells and the clinical side of their uses.
- More difficult, the cellular expression models have to be justified. It is well known that specific knockdown approaches can have broad effects on the cellular homeostasis. It is briefly stated that the results obtained here would demonstrate that the model was suitable for the network analysis; however, a more critical review taking the predictive nature of results into account would be needed.
Response: we agree that knock-down approaches require rigorous controls, in order to avoid the risk of unspecific cellular responses or observations. This has been carefully evaluated during the review process of our two base publications (doi: 10.1016/j.jid.2022.07.020 and doi: 10.1038/s41551-019-0464-6).
As detailed in these two base publications, for both KLF4 and MXD4/MAD4 studies, at least two shRNA sequences were used for stable lentiviral-mediated down-modulation target transcripts. In addition, at least two different siRNAs were used for transient down-modulation. Demonstrations also included experiments involving over-expression.
- An innovative part is the profiling of lncRNAs alongside protein-coding RNA signatures. These data identify new strategies for experimental studies and require such validations. It should be clarified to which extent they can be relevant: Linking lncRNA signatures to functionally annotated protein-coding RNA pathways seems to be promising; given that the protein-coding RNA networks have been developed from an experimental 'stemness' model, the validity of this alignment should be more critically discussed.
Response: accordingly, the following comment has been added to the discussion (page 25, lines 589-595).
Regarding future prospects for new candidates (lncRNA or other), the same strategy used for KLF4 and MXD4 studies will be implemented using functional genomics tools suitable for targeted known-down or over-expression, including shRNAs, small-interfering RNAs, antisense oligonucleotides, and over-expression systems. This experimental phase will be necessary to move from the stage of prospective identification to that of demonstrated biological functions, as we did for the KLF4 and MXD4/MAD4 transcription factors.
Minor point:
- The first paragraph of the Results section (Cellular models) refers to previous publications of the authors. Their generation is not done and not described here and could be more suitably arranged in the text.
Response: generation of the cellular model and its main characteristics are summarized in the paragraph 2.1. (Results):
The cellular model of human holoclone keratinocytes was used to decipher the molecular network responsible for the regulation of human epidermal stem and precursor cell fate. These cells correspond to the clonal progeny of single keratinocyte stem cells. They have been functionally characterized ex vivo by their growth potential exceeding 100 population doublings in long-term culture and their capacity for three-dimensional (3D) epidermis reconstruction. Additionally, holoclone keratinocytes have the potential for long-term in vivo grafting. The functional genomics approach designed to explore the regulatory roles of KLF4 and MXD4/MAD4 was based on the generation of stable targeted knock-down (KD) cellular contexts. These contexts were then used for comparative studies of KD cells versus their wild-type (WT) equivalent. Lentiviral vectors driving expression of shRNAs directed against the KLF4 or MXD4 transcripts were used to transduce holoclone keratinocytes and obtain stable KD cells. Comparisons were performed versus cells transduced with a control vector. The KLF4KD and the MXD4KD strategies converged to promote an augmented ex vivo cellular expansion, associated with improved maintenance of stem and precursor cell clone-forming efficiency, together with preservation of potential for epidermis generation
To direct readers more clearly to the source data and corresponding original description, the following sentence has been added at the beginning of the paragraph (page 2, line 91):
“As previously described in our original articles [16,19] …”

Reviewer 3 Report
Comments and Suggestions for Authors
Excellent study on stemness of keratinocytes and holoclones. Great data analysis. Very robust. Well written. Useful figures.
Minor comments only:
Any common integration/cross-talk of the EGF, BDNF, and WNT pathways? It would be better to be more specific.
TGFB/BMP signaling could be associated or resultant of a Th2 shift. How were the other Th2 main genes trending?
Given TGFB's roles seem ubiquitous and it comes back often, discussion section on TGFB pathway should be enhanced.
Would add a summary figure - much simpler than the network ones.
Author Response
Manuscript ID: ijms-2887060
Point-by-point responses to Reviewer’s comments and suggestions:
Black font: reviewer’s comments and suggestions.
Blue font: author’s responses.
Red font: addition in the manuscript.
Review Report 3
Comments and Suggestions for Authors
Excellent study on stemness of keratinocytes and holoclones. Great data analysis. Very robust. Well written. Useful figures.
Response: We thank reviewer 3 for this very positive evaluation of our manuscript, and for the constructive comments and suggestions.
Minor comments only:
Any common integration/cross-talk of the EGF, BDNF, and WNT pathways? It would be better to be more specific.
Response: At present, we cannot provide a clear response as no study has directly addressed this point in interfollicular keratinocytes.
In addition to the ‘summary figure’ (last reviewer 3 request), the following discussion point has been added in the revised manuscript (page 23, lines 496-504).
Regarding the possible cross-talk between the EGF, BDNF, and WNT pathways (see figure 12), as suggested by our in silico analysis, no published data is available to directly address this point in interfollicular keratinocytes. It is worth noting that EGFR has been shown to play a critical role in attenuating Wnt/β-catenin signaling during postnatal mouse hair follicle development by directly acting on stem cell-specific markers such as SOX9 and NFATc1 (Tripurani et al., 2018, doi: 10.1091/ mbc.E18-08-0488). A comparable mechanism cannot therefore be excluded in interfollicular epidermis keratinocytes. However, there is no available data to confirm the contribution of BDNF in this potential cross-talk among keratinocytes.
TGFB/BMP signaling could be associated or resultant of a Th2 shift. How were the other Th2 main genes trending? (lymphocyte T helper 2)
Response: concerning the question of a relationship between keratinocytes and T lymphocyte activation/recruitment, studies have been performed in the context of skin inflammation or cutaneous diseases such as atopic dermatitis or psoriasis. Influence of T helper lymphocytes on the modulation of the keratinocyte immaturity degree is not well understood. Also, the ‘interferon signaling pathways’ depicted in figures 7 and 8 are modulated in an opposite manner in our two cellular models (up-modulated in KLF4KD and down-modulated in MXD4KD). As raised by several data presented in the manuscript, transcript modulation can differ depending on the cellular model. This point is mentioned in the revised manuscript (page 24, lines 535-538).
Of note, some transcript groups and pathways appeared modulated in an opposite manner in our two cellular models (for example transcripts assigned to ‘interferon signaling pathways’), which can be interpreted as the result of specificities distinguishing the regulatory networks of KLF4 and MXD4/MAD4.
Given TGFB's roles seem ubiquitous and it comes back often, discussion section on TGFB pathway should be enhanced.
Response: discussion section on TGFB's documented roles in the control of progenitor cell immaturity / differentiation orientation has been enhanced, as follows (page 23, lines 490-496):
Moreover, TGFB signaling inhibition associated with ROCK signaling inhibition, synergistically supports long-term expansion of epithelial precursor cells from prostate, bronchial tissue and skin (Zhang et al., 2018, doi: doi.org/10.1016/j.celrep.2018.09.072). In addition, Activin/TGFB signaling is also involved in the quiescence of hair follicle keratinocytes, in mice (Kadaja et al., 2014, doi: 10.1101/gad.233247.113). Interestingly, the key role of TGFB signaling in the regulation of progenitor cell fate has been also documented in the human hematopoietic system (Fortunel et al., 2000, doi.org/10.1182/blood.V96.6.2022), notably in the control of cell-cycling and self-renewal (Batard et al., 2000, doi: 10.1242/jcs.113.3.383), and in the equilibrium between immaturity and erythroid differentiation (Tanabe et al., 2022, doi: 10.1038/s41375-021-01463-3).
Would add a summary figure - much simpler than the network ones.
Response: we agree that a final ‘more simple’ figure highlighting a main take-home-message will enhances the clarity of the delivered message. The following figure has been added, highlighting a pathway involving TGFB / WNT / growth factors as a candidate machinery of keratinocyte immaturity-stemness control (pages 24, lines 516-522).
Figure 12. LncRNAs potentially involved in the immaturity-stemness regulation processes in the human keratinocyte lineage. Schematic representation of the hypothetical lncRNAs participation on TGFB, PI3K/MAPK and WNT pathways. Activator and repressor pathway proteins are green and purple respectively; lncRNAs are loop RNA structures and involved miRNAs are small dark blue structures. Confirmed molecule interactions are black arrows, co-expression interactions are orange arrows.
